# Identification of Serum Oxylipins Associated with the Development of Coronary Artery Disease: A Nested Case-Control Study

**DOI:** 10.3390/metabo12060495

**Published:** 2022-05-30

**Authors:** Kuang-Mao Chiang, Jia-Fu Chen, Chin-An Yang, Lili Xiu, Hsin-Chou Yang, Lie-Fen Shyur, Wen-Harn Pan

**Affiliations:** 1Institute of Biomedical Sciences, Academia Sinica, Taipei 11529, Taiwan; kuangmao@ibms.sinica.edu.tw; 2Institute of Public Health, School of Medicine, National Yang-Ming Chiao Tung University, Taipei 11221, Taiwan; zaicross@hotmail.com.tw; 3Department of Biochemical Science and Technology, National Taiwan University, Taipei 10617, Taiwan; chin-an.yang@emory.edu; 4Institute of Population Health Sciences, National Health Research Institutes, Miaoli 35053, Taiwan; xll_200288@hotmail.com; 5Institute of Statistical Science, Academia Sinica, Taipei 11529, Taiwan; hsinchou@stat.sinica.edu.tw; 6Agricultural Biotechnology Research Center, Academia Sinica, Taipei 11529, Taiwan; lfshyur@ccvax.sinica.edu.tw

**Keywords:** fried oil, oxylipin, linoleic acid, coronary artery disease, NAHSIT, metabolomics

## Abstract

Coronary artery disease (CAD) is among the leading causes of death globally. The American Heart Association recommends that people should consume more PUFA-rich plant foods to replace SFA-rich ones to lower serum cholesterol and prevent CAD. However, PUFA may be susceptible to oxidation and generate oxidized products such as oxylipins. In this study, we investigated whether the blood oxylipin profile is associated with the risk of developing CAD and whether including identified oxylipins may improve the predictability of CAD risk. We designed a nested case-control study with 77 cases and 148 matched controls from a 10-year follow-up of the Nutrition and Health Survey in a Taiwanese cohort of 720 people aged 50 to 70. A panel of 46 oxylipins was measured for baseline serum samples. We discovered four oxylipins associated with CAD risk. 13-oxo-ODE, which has been previously found in formed plagues, was positively associated with CAD (OR = 5.02, 95%CI = 0.85 to 15.6). PGE2/PGD2, previously shown to increase cardiac output, was inversely associated (OR = 0.16, 95%CI = 0.06 to 0.42). 15-deoxy-PGJ2, with anti-inflammatory and anti-apoptosis effects on cardiomyocytes (OR = 0.26, 95%CI = 0.09 to 0.76), and 5-HETE, which was associated with inflammation (OR = 0.28, 95%CI = 0.10 to 0.78), were also negatively associated as protective factors. Adding these four oxylipins to the traditional risk prediction model significantly improved CAD prediction.

## 1. Introduction

Coronary artery disease (CAD) is the most common type of heart disease and cardiovascular disease (CVD) in Taiwan and in the world. To prevent CVD, the American Heart Association (AHA) recommends that people consume more polyunsaturated fatty acid (PUFA)-rich plant foods to replace saturated fatty acid (SFA)-rich ones to lower serum cholesterol and prevent CVD. However, PUFA may be oxidized in vivo or in vitro and generate oxidized compounds [1]. When PUFAs in foods are exposed to air and moisture in high-temperature conditions such as the frying process, a wide array of oxylipins1 is generated. Our previous study showed that serum metabolite profiles correspond to the types of cooking oils consumed [2]. In the human body, three major types of oxidases also oxidize PUFA into a whole spectrum of oxylipins. The various roles of oxylipins have been implicated in multiple physiological functions, including vasoconstriction, vasodilation, inflammation, clotting, and atherosclerosis [3]. Some animal studies have shown that certain PUFA metabolites may be associated with the risk of cardiovascular diseases and cancers [3]. It has also been observed in human studies that some significant cardiovascular diseases, such as hyperlipidemia, hypertension, diabetes, thrombosis, and hemostasis, have been linked to oxylipins [4]. In addition, a recent 5-year follow-up study that measured 39 oxylipins in plasma samples from 74 CAD patients found that the concentrations of six oxylipins in the plasma were associated with CAD severity and survival [5].

On the other hand, some prospective studies have reported a positive and significant association between coronary artery disease (CAD) and the frequency of consuming fried foods. A prospective study that used the combined data of 70,842 U.S. female subjects from the Nurses’ Health Study (1984–2010) and 40,789 U.S. male subjects from the Health Professionals Follow-Up Study (1986–2010) showed that frequent fried-food consumption was significantly associated with the risk of incident CAD and type 2 diabetes [6]. These associations were mediated by body weight and comorbid hypertension and hypercholesterolemia. However, another 11-year follow-up study that enrolled 40,757 adults from a Spanish cohort of European Prospective Investigation into Cancer and Nutrition (EPIC) demonstrated that in Spain, a Mediterranean country where olive or sunflower oil is used for frying, the consumption of fried foods was not associated with CAD [7]. This inconsistent result may be in part due to the different fatty acid compositions of the oils used in the United States and Spain. The Mediterranean diet has demonstrated protective effects against cardiovascular disease [8].

Although it is in general considered unhealthy to eat deep-fried foods due to their high caloric content, their effects on other health dimensions have not been well studied. Since few epidemiological studies have examined the association between oxylipins and CAD, we set out to explore the association in humans and investigate whether adding discovered oxylipins to the traditional risk assessment model would increase the predictability of CAD or not.

## 2. Results

Table 1 shows the results of the baseline demographics and biochemical indicators of 77 CAD cases and 134 age-, sex-, residential area-, and season of interview-matched controls. The average age of the cases and controls was 61.7 ± 5.1 years. Around half (51.2%) of the participants were male. Between the two groups, only the mean concentration of the total triacylglycerol at the baseline was significantly higher in the incident CAD cases than in the controls (*p* = 0.02).

### 2.1. Conditional Logistic Regression Analysis

Four oxylipins (i.e., 13-oxo-ODE, 5-HETE, PGD2/PGE2, and 15-deoxy-PGJ2 (15d-PGJ2)) were significantly associated with incident CAD from the results of conditional logistic regression with covariate adjustment (Table 2). Appendix A is the boxplot of these four oxylipins. The association between 5-HETE and CAD was not significant if we only adjusted for smoking, alcohol consumption, education, and experimental batch effects, and not clinical CAD risk factors (see Appendix A). The studied subjects were divided into tertiles according to oxylipin concentration level (ppm). The first tertile, with the lowest range of oxylipins, was treated as the reference group. Among the four oxylipins, PGD2/PGE2, 15d-PGJ2, and 5-HETE were inversely associated with CAD events. For PGD2/PGE2, the odds ratios (OR) of the middle tertile and the highest tertile groups were significantly lower than 1 (OR and 95% confidence interval [CI] = 0.33 [0.15, 0.71] and 0.27 [0.12, 0.60], respectively) with a significant p for trend (*p* = 0.001). For 15d-PGJ2, the OR of the highest tertile group was significantly lower than 1 (OR and 95% CI = 0.33 [0.14, 0.79]) as well. The *p*-value for trend was also significant (*p* for trend = 0.01). For 5-HETE, the OR of the middle tertile group was significantly lower than the lowest tertile group (OR and 95% CI = 0.43 [0.19, 0.96]). The OR of the highest tertile group was also lower than the lowest group (OR and 95% CI = 0.61 [0.27, 1.35]), but did not reach the statistical significance level.

For 13-oxo-ODE, the OR of the middle tertile group was significantly higher than 1 (OR and 95% CI = 2.48 [1.12, 5.48]). The highest tertile group also had a greater risk than the lowest tertile group to develop CAD, but the corresponding OR (OR and 95% CI = 1.81 [0.85, 3.84]) did not reach the statistical significance level. 13-oxo-ODE appears to have had a risky effect, whereas 5-HETE, PGD2/PGE2, and 15d-PGJ2 were protective. We further adjusted the education level in model 2. The results are similar with very slight changes in the ORs and 95% CIs (Table 2).

### 2.2. Multivariate Logistic Regression

We further used a multivariate logistic regression model to find the independent predictors. These results show that 13-oxo-ODE remained the independent risk effect, whereas two (PGD2/PGE2, 15d-PGJ2) were independent protective factors (Table 3). For 5-HETE, the OR of the middle tertile group was significantly lower than the reference, but the highest group did not reach statistical significance. The trend test of 5-HETE was also not significant.

### 2.3. ROC Curve Analysis

Furthermore, we used the ROC curve analysis to examine the predictability of the incidence of CAD, as shown as the area under the curve (AUC) in Appendix A and Figure 1. Compared to using only the traditional CAD predictors (age, sex, status of hypertension, diabetes, hypertriglyceridemia, hypercholesterolemia, smoking, drinking, and education), when we added each single identified oxylipin to the prediction model, the AUC increased from 0.63 to 0.71 for PGE2/PGD2, to 0.68 for 15-deoxyPGJ2, to 0.66 for 13-oxoODE, and to 0.65 for 5-HETE. However, these differences were not statistically significant except for PGE2/PGD2 (*p* = 0.015). We further added the oxylipins to the prediction model one by one according to the predictability of these four oxylipins. As shown in Appendix A, with the addition of more oxylipins, the AUC also increased. The predictability significantly increased from 0.63 to 0.76 after adding the four oxylipins (*p*-value < 0.001).

### 2.4. Sensitivity Analysis

We performed a sensitivity analysis using the LASSO method to select the predictors to check whether the chosen metabolites could still be selected in the Lasso-developed model. We started by entering traditional risk factors (age, sex, smoking, drinking, education hypertension, diabetes, hypertriglyceridemia, hypercholesterolemia, and experimental batch) into the model. After filtering by the LASSO regression model, hypertriglyceridemia and diabetes were selected into the prediction model (Appendix A) and the R-square of this model was 0.017. We then further added the four identified metabolites into the model. The LASSO method identified all four identified metabolites in the final model (Appendix A). We noticed that some predictors, such as drinking, education, hypertension, and hypercholesterolemia, were also selected in the prediction model after adding these oxylipin metabolites. The R-square of this final prediction model was improved from 0.017 to 0.13.

## 3. Discussion

This study is the first prospective one using a targeted metabolomics approach to relate oxylipins with CAD risk in the Chinese population. We found three oxylipins (PGD2/PGE2, 15d-PGJ2, and 5-HETE) were inversely and one (13-oxo-ODE) was positively associated with CAD development. Further, in multivariate logistic regression models, 13-oxo-ODE, PGD2/PGE2, and 15-deoxy-PGJ2 demonstrated independent effects on CAD risk. Finally, by ROC curve analysis, we found that the prediction power of the oxylipins-enriched model was significantly higher than that with only traditional risk factors. Further confirmation study is highly warranted.

The magnitude of the effects of 13-oxo-ODE is not trivial. People in the highest tertile experienced more than five times the CAD risk than those in the lowest tertile. 13-oxo-ODE (13-oxo-octadecadienoic acid) is one of the oxidized linoleic acid metabolites (OXLAMs). Linoleic acid (LA) is the most consumed polyunsaturated fatty acid in human diets, a major component of human tissues, and the direct precursor to 13-oxoODE [9]. LA oxidation can proceed enzymatically or non-enzymatically [10]. OXLAMs have been reported to play an important role in foam cell formation and atherosclerosis plaques [11]. A previous study also found that circulating OXLAMs were elevated in non-alcoholic steatohepatitis (NASH) [12]. Both conditions are known as intermediates leading to the development of CAD. It is crucial to understand the health impacts of OXLAMs and the facilitatory or inhibitory mechanisms of OXLAM formation.

Prostaglandin D2 and E2 (PGD2 and PGE2) have very similar structures. These two oxylipins are both produced from arachidonic acid but through different pathways, and arachidonic acid is an enzymatic downstream product of LA. We could not differentiate these two using the LC-MS system due to their structural similarity. Previous animal studies have shown that PGE2 has a protective effect against myocardium and myocardial ischemia [13], and PGD2 has a protective effect against ischemia–reperfusion injury of isolated rat hearts [14].

As a downstream metabolite of PGD2 [15], 15d-PGJ2 has recently been identified as an anti-inflammatory substance. A previous study showed that 15d-PGJ2 can modify the transcriptional activation of many proinflammatory genes, such as down-regulating IL-6 production and up-regulating IL-8 production via the suppression of the NF-*κ*B and MAPK signaling pathways [14]. In the multivariate logistic regression analysis, all of these three prostaglandins showed independent protective effects. High level (upper tertile) PGD2/PGE2 or 15d-PGJ2 may respectively reduce CAD risk by about 70–80% in comparison to the lowest tertile group.

Our research has several strengths. With a validated targeted metabolomics approach, we measured 46 known oxylipins in one run and provided accurate peak detection and quantification. In addition, the baseline data from the representative Taiwanese nutrition and health survey database together with the national health insurance data for event identification provide an opportunity to make causal inference. This study also has some limitations. The sample size of this nested-case control study was modest. A further large-scale prospective study should be carried out to confirm our findings. Oxylipins are correlated with each other. Whether the effect of 5-HETE is masked by other oxylipins should not be overlooked.

In summary, our study identified four oxylipins associated with CAD risk. Among them, one had independent risk and three were protective factors. Adding these metabolites to a traditional model may improve CAD event prediction. More research is needed to confirm our findings and to understand the effects and mechanisms of the identified oxylipins in CAD pathogenesis.

## 4. Materials and Methods

### 4.1. Research Design and Participants

In this study, we designed a nested case-control study, taking advantage of the data and bio-specimens from the Nutrition and Health Survey in Taiwan (NAHSIT) 1993–1996 [16]. A total of 720 participants aged 50 to 70 years at the baseline were included in this prospective investigation. The cohort was followed-up with the Taiwan National Health Insurance (NHI) database until 31 December 2002. The NHI program provides lifelong medical coverage for all Taiwanese citizens with minimal deductibles. In this study, the new incident CAD cases were identified when the following codes were documented: 410 to 414 of the ICD-9 CM codes and A270 and A279 of the A codes. The participants were defined as CAD cases when the participant’s hospitalization or death record had any of the above codes. From those 720 participants who did not have any CAD codes prior to the baseline, 77 people developed CAD during the follow-up period (median year of onset = 5.16 years). Up to 2 controls were matched to each case with respect to age (±2.5 years), sex, residential area, and season of interview. A total of 138 matched controls who did not develop CAD before 31 December 2002 were chosen. Fasting serum samples collected at the baseline from the 77 cases and 138 controls were used to carry out metabolomics profiling (Figure 2). Other baseline information used included age, sex, education, smoking, drinking, diastolic blood pressure (DBP), systolic blood pressure (SBP), fasting glucose (FG), triglyceride (TG), VLDL-C, LDL-C, HDL-C, total cholesterols (TC), and medication.

### 4.2. Sample Storage and Preparation

The fasting serum samples collected at the baseline were frozen at −80 °C. Following the oxylipin extraction protocol developed by Maria Karmella Apaya et al. (2016), 300 μL of serum was added into a 2 mL tube, followed by the addition of 900 μL of extraction solvent (CHCl_3_: MeOH = 2:1, 5 mg BHT/per 200 mL CHCl_3_ 200 mg TPP/per 200 mL CHCl_3_) and 7.5 μL of internal standard solution. The internal standard solution, which consisted of six commercial deuterium-substituted oxylipins (9-HODE-d4, 5-HETE-d8, PGE2-d4, 14,15-EET-d11, EPA-d5, and DHA-d5) in the methanol solvent, was used to check retention time shifts and peak detection. The mixture was stirred by vortex for 30 s and then kept at −20 °C for 20 min to precipitate out the proteins. After precipitation, the mixture was centrifuged at 4 °C and 13,000× *g* rpm for 10 min, and then we transferred the supernatant to a 1.5 mL tube. Every sample was extracted twice and the combined supernatant was dried with nitrogen gas. The dried analytes were dissolved in 75 μL of MeOH before being injected into the LC/MS system for metabolomics measurements [17].

### 4.3. Oxylipins Measurement by Liquid Chromatography–Electrospray Ionization Tandem Mass Spectrometry (LC-ESI-MS/MS)

The system used for metabolomic profiling was an ACQUITY Ultra Performance Liquid Chromatography (UPLC) system (Waters Corp., Milford, MA, USA) coupled with a heated electrospray ionization (HESI) source of a TSQ Quantum Access Max (Thermo Fisher Scientific, San Jose, CA, USA) triple quadrupole mass spectrometer. A plasma sample of 10 μL was injected into an ACQUITY UPLC BEH C18 column (particle size of 1.7 μm, 2.1 × 100 mm, Waters, Milford, MA) and separated at 400 μL/min flow rate using a 25 min gradient for analysis. The mobile phase A was 0.1% NH_4_OH in water and the mobile phase B was 0.1% NH_4_OH in MeOH. The gradient elution program was as follows: 0–1 min, 8% B; 1–15 min, linear gradient from 8 to 80% B; 15–18 min, 80% B; 18–18.5 min, linear gradient from 80 to 100% B; 18.5–22 min, 100% B; 22–22.5 min, linear gradient from 100 to 8% B, and kept in 8% B for 2.5 min. The LC-MS was operated in the negative multiple reaction monitoring (MRM) mode. Precursor and the most abundant fragment ions for the quantification of each analyte were obtained by scanning individual analyte standards separately in the MS full-scan mode and product ion scan mode. The source parameters and collision energy of each analyte were optimized by direct infusion. The MS operating parameters were set as follows: spray voltage, 2.7 kV; vaporizer temperature, 300 °C; ion transfer capillary temperature, 270 °C; sheath gas (nitrogen), 40 Arb; auxiliary gas (nitrogen), 10 Arb; collision gas (argon) pressure, 1 mTorr. To solve the matrix effect, we added 10 μL of all plasma samples to the standard solution to check retention time shifts and peak intensities. Each serum sample was measured three times (three technical replicates) to detect system variability.

### 4.4. Data Processing

Using the targeted metabolomics approach, we analyzed 46 oxylipins with established protocol in this study (Appendix A). Using ThermoXcalibur 2.1 SP1 software (Thermo Fisher Scientific Inc., Waltham, MA, USA), we viewed the chromatogram and detected the mass spectral peaks and waveform. The peak area quantification was performed by LCQuan 2.6.1 software (Thermo Scientific). Some missing values caused by shifted peaks or deconvolution of the overlapping need to be substituted [18]. To avoid the false findings caused by excessive missing values, we applied the substitution methods suggested by previous studies and optimized the peak intensity value from three replicates of each sample [19]. After substitution, we checked the missing rate of each oxylipin and removed the ones with missing rates above 50% (Resolvin D1, 10,17-DiHOME, Maresin, and 6-keto-PGF1a).

Statistical Metabolomics Analysis—An R Tool (SMART) v1.2 [20] was used for data normalization, quality control, and batch effect detection. The peak area intensities were normalized by the Pareto-scaling method, and then we checked the consistency among the three replicates of each sample by the quality control process. After the above quality control process, we used the median of three replicates or the mean of two replicates to represent the intensity of each oxylipin peak in each sample. After the data cleaning (h > 0.083, Appendix A), four controls were removed from the analysis with poor experimental data quality. In total, we had complete information on 44 oxylipins in 77 cases and 134 controls in this study (Figure 2). The original concentrations of these 42 oxylipins are shown in Appendix A. We carried out the principal component analysis (PCA) method to detect batch effects and found two PCs. The two PC variables have been adjusted in all the statistical models.

### 4.5. Statistical Analysis

The descriptive data comparisons between the case and the control groups were performed using Fisher’s exact test or *t*-test. Since the age, sex, place of residence, and season at the time of the interview were matched, we used conditional logistic regression models with adjustment for covariates to search for oxylipins that may be potentially associated with the risk of developing CAD. In this study, two conditional logistic regression models were performed. In model 1, the covariates adjusted include hypertension, diabetes, hypertriglyceridemia, hypercholesterolemia, alcohol consumption, smoking, and batch effects identified for metabolomics profiling. In model 2, we further adjusted for the education level in addition to the covariates adjusted in model 1.

We further used the multivariate logistic regression models to check whether the effect of each identified oxylipin was independent of other oxylipins. Lastly, we used the receiver operating characteristic (ROC) curve method to estimate and compare the prediction powers between the traditional model and that enriched with oxylipins.

To avoid overfitting, we also performed a sensitivity analysis using the LASSO method [21] to select the predictors to check whether the identified metabolites would be retained in the LASSO model and to compare the explanatory power of the prediction models with two methods. The LASSO method is a shrinkage regression technique using L1 regularization and is designed for high-dimensional data. This algorithm shrinks the coefficients of noninfluential predictors to zero and, thus, excludes them from the final model. This technique has been widely used in both machine learning and clinical practice. All statistical analyses were performed using SAS 9.4 software (SAS Institute Inc., Cary, NC, USA).

## Figures and Tables

**Figure 1 metabolites-12-00495-f001:**
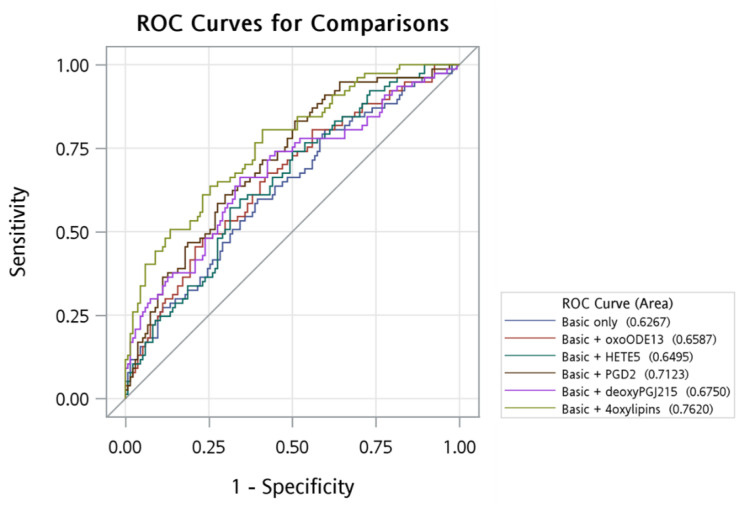
ROC curve of basic variables and oxylipins.

**Figure 2 metabolites-12-00495-f002:**
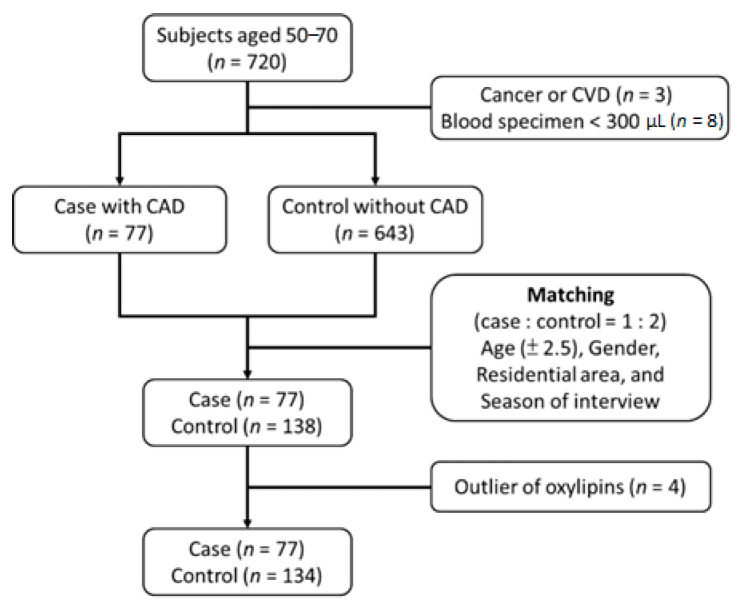
Study design flow chart of the nested case-control study on coronary artery disease.

**Table 1 metabolites-12-00495-t001:** Baseline characteristics of participants by coronary artery disease (CAD) status.

Characteristics	All Participants	Case	Control	*p*-Value
	(*n* = 211)	(*n* = 77)	(*n* = 134)	
**Gender (%)**				Matched
Male	108 (51.2)	39 (50.6)	69 (51.5)	
Female	103 (48.8)	38 (49.3)	65 (48.5)	
**Age** **(year** **)**	61.7 ± 5.1	61.7 ± 5.1	61.6 ± 5.1	Matched
**Education level**				0.47
Less than elementary school	47 (21.8)	17 (22.1)	30 (22.4)	
Elementary school	108 (51.2)	36 (45.7)	72 (53.7)	
More than elementary school	56 (27.0)	23 (31.3)	32 (23.9)	
**Hypertension**	104 (48.8)	37 (48.1)	67 (50.0)	0.88
**Stroke**	3 (1.4)	2 (2.6)	1 (0.7)	0.55
**Diabetes Mellitus**	44 (20.5)	20 (25.9)	24 (17.4)	0.21
**Kidney Disease**	8 (3.8)	1 (1.3)	7 (5.2)	0.26
**Stone**	6 (10)	0 (0)	6 (4.5)	0.08
**Smoking**				0.72
Non-smoker	116 (54.9)	42 (54.5)	74 (55.2)	
Current smoker	43 (20.4)	15 (19.4)	28 (20.9)	
Quit smoking	17 (8.1)	8 (10.4)	9 (6.7)	
Ever smoked	10 (4.8)	2 (2.8)	8 (6.0)	
**Drinking**				0.76
Never	48 (22.7)	16 (20.7)	32 (23.8)	
Below 1/week	49 (23.2)	20 (25.9)	29 (21.6)	
Above 1/week	35 (16.6)	10 (13.1)	25 (18.6)	
Quit	24 (11.4)	11 (14.3)	13 (9.7)	
Ever	32 (15.2)	11 (14.3)	21 (15.7)	
**Waist circumference (cm)**	82.3 ± 9.9	83.4 ± 9.4	81.6 ± 10.2	0.22
**BMI, (kg/m^2^)**	24.5 ± 3.8	24.8 ± 3.5	24.3 ± 3.9	0.37
**LDL-C (mg/dl)**	122.3 ± 42.5	123.5 ± 45.6	121.7 ± 40.8	0.76
**HDL-C (mg/dl)**	53.7 ± 17.3	51.8 ± 16.4	54.8 ± 17.9	0.23
**Total cholesterol (mg/dl)**	205.2 ± 47.1	210.4 ± 44.8	202.3 ± 48.2	0.29
**Total triacylglycerol (mg/dl)**	148.2 ± 125.3	175.1 ± 164.4	134.2 ± 95.5	0.02 *
**Blood sugar (mg/dl)**	99.1 ± 38.9	101.2 ± 40.1	98.0 ± 38.2	0.56
**Medication**				
Anti-hypertensive	34 (16.3)	16 (20.8)	18 (13.4)	0.17
Anti-diabetic	19 (8.8)	10 (13.0)	9 (6.5)	0.13
Lipid-lowering agent	5 (2.3)	2 (2.6)	3 (2.2)	-

* *p*-value < 0.05.

**Table 2 metabolites-12-00495-t002:** Adjusted odds ratio (OR) and 95% confidence interval (CI) for CAD associated with tertile of each oxylipin.

Oxylipins	Borderline	Number of	Model 1		Model 2	
	(ppm)	Cases/Controls	OR (95% CI)	*p* for Trend	OR (95% CI)	*p* for Trend
**13-oxo-ODE**				0.1		0.09
Low tertile	<0.0055	18/52	Ref.		Ref.	
Middle tertile	0.0055–0.006	31/40	2.48 (1.12, 5.48) *		2.75 (1.20, 6.26) *	
High tertile	≥0.006	28/42	1.81 (0.85, 3.84)		1.86 (0.86, 4.00)	
**5-HETE**				0.17		0.2
Low tertile	<0.0173	30/40	Ref.		Ref.	
Middle tertile	0.0173–0.0180	20/51	0.43 (0.19, 0.96) *		0.44 (0.19, 0.98) *	
High tertile	≥0.0180	27/43	0.61 (0.27, 1.35)		0.63 (0.28, 1,41)	
**PGD2/PGE2**				0.001 **		0.001 **
Low tertile	<0.001001	37/33	Ref.		Ref.	
Middle tertile	0.001001–0.001005	20/51	0.33 (0.15, 0.71) *		0.34 (0.15, 0.72) *	
High tertile	≥0.001005	20/50	0.27 (0.12, 0.60) *		0.28 (0.13, 0.61) *	
**15-deoxy-PGJ2**				0.01 *		0.01 *
Low tertile	<0.001002	29/41	Ref.		Ref.	
Middle tertile	0.001002–0.00101	30/41	0.92 (0.44, 1.91)		1.00 (0.48, 2.09)	
High tertile	≥0.00101	18/52	0.33 (0.14, 0.79) *		0.32 (0.13, 0.78) *	

Model 1: Adjusted for hypertension, diabetes mellitus, hypertriglyceridemia, hypercholesterolemia, drinking, smoking, and experimental batch effects. Model 2: Adjusted for above variables and education level. * *p*-value < 0.05, ** *p*-value < 0.01. Ref. = reference group.

**Table 3 metabolites-12-00495-t003:** Independent adjusted OR and 95% CI for CAD associated with tertile of each oxylipin.

Oxylipins	Borderline	Number of	Model 1		Model 2	
	(ppm)	Cases/Controls	OR (95% CI)	*p* for Trend	OR (95% CI)	*p* for Trend
**13-oxo-ODE**				0.003 **		0.003 **
Low tertile	<0.0055	18/52	Ref.		Ref.	
Middle tertile	0.0055–0.006	31/40	4.36 (1.53, 12.4) **		5.52 (1.75, 17.4) **	
High tertile	≥0.006	28/42	5.02 (1.85, 15.6) **		5.51 (1.68, 18.0) **	
**5-HETE**				0.15		0.18
Low tertile	<0.0173	30/40	Ref.		Ref.	
Middle tertile	0.0173–0.0180	20/51	0.27 (0.09, 0.74) **		0.28 (0.10, 0.78) **	
High tertile	≥0.0180	27/43	0.47 (0.15, 1.43)		0.52 (0.17, 1.61)	
**PGD2/PGE2**				0.0004 **		0.001 **
Low tertile	<0.001001	37/33	Ref.		Ref.	
Middle tertile	0.001001–0.001005	20/51	0.26 (0.10, 0.66) **		0.25 (0.09, 0.64) **	
High tertile	≥0.001005	20/50	0.16 (0.06, 0.42) **		0.15 (0.05, 0.40) **	
**15-deoxy-PGJ2**				0.01 *		0.01 *
Low tertile	<0.001002	29/41	Ref.		Ref.	
Middle tertile	0.001002–0.00101	30/41	0.58 (0.23, 1.47)		0.62 (0.24, 1.59)	
High tertile	≥0.00101	18/52	0.33 (0.09, 0.76) **		0.23 (0.07, 0.69) **	

Model 1: Adjusted for hypertension, diabetes mellitus, hypertriglyceridemia, hypercholesterolemia, drinking, smoking, experimental batch effects, and these oxylipins. Model 2: Adjusted for above variables and education level. * *p*-value < 0.05, ** *p*-value < 0.01. Ref. = reference group.

## Data Availability

All data are available from the authors upon reasonable request.

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
