# Peer review of "Identification of Serum Oxylipins Associated with the Development of Coronary Artery Disease: A Nested Case-Control Study"

_metabolites, 2022, doi:10.3390/metabo12060495_

Round 1

Reviewer 1 Report

The authors reported in this paper that several oxylipins are associated with coronary artery disease either positively or negatively. They performed a nested case-control study from 720 subjects and followed the subjects for 7-10 years. This is an interesting study to search for novel markers for coronary artery disease, and their conclusion may add new candidates in the list of risk markers. However, there are several critical concerns to be answered.

Minor points

  1. Although lipidomic analysis using LC-MS/MS has been extensively performed by many groups, the methodology and data analysis are still not popular to many of the readers. The authors should provide data in detailed enough to explain the whole picture of this analysis. Firstly, the list of detection conditions of all oxylipins tested (retention time, m/z of molecular ions and fragment ions used for MRM detection) should be shown. Secondly, the amounts of all the oxylipin molecules detected should be shown in a Table of Figure. The information shown in the manuscript are only the borderline concentrations used in the statistical analysis of selected oxylipins, which are totally insufficient to understand the whole view of the comprehensive analysis of a series of oxylipins.
  2. The list of 46 oxylipins might have been prepared as Appendix 1 (Section 2.4.), however, this list is missing in the PDF file available for this reviewer.
  3. In this study the authors selected 148 subjects from the cohort of 720 participants to make a control group in which the baseline parameters are well matched with the case group. I am afraid that the two groups are not well-matched. TG levels are significantly different between the case and control groups, and I think this parameter is critically important for studying lipid metabolites.
  4. Why model 2 was investigated by adjusting educational levels? Table 1 shows there is little difference in educational levels between case and control groups. I do not understand why model 2 is necessary. In addition, the number of subjects in model 2, namely after adjusting educational levels, is not indicated.
  5. Figure 2. I cannot follow how the ROC curve was calculated. The blue line in the figure represents the ROC curve of “basic” parameters which are 10 traditional parameters, and the red line the data of 10 traditional plus 3 oxyline parameters. How can you calculate a specific parameter to evaluate the probability of the disease from these multiple parameters? Did you show the equation to calculate the specific parameters?

Minor points

  1. The authors did not site previous studies related to oxylipins in cardiovascular diseases.
  2. The name of AHA is misspelled.

Reviewer 2 Report

Chiang and colleagues analyzed the association between oxylipins and the risk of coronary artery disease in the Nutrition and Health Survey in Taiwan (NAHSIT) cohort study. Despite the small sample size that the authors admitted as one of the limitations of the study, they found four oxylipins that were associated with the risk of coronary artery disease (CAD). Adding these four oxylipins to the basic prediction model based on the conventional risk factors of CAD improve the prediction model. Here are some feedbacks that the authors can use to improve the manuscript.

General comments:

  1. Introduction: The link between fried foods and oxylipins that become the focus of the study seems missing. Is there perhaps any research showing that fried foods contain oxylipins, and if oxylipins have been previously associated with CAD? It would be great if a mediation study showing how oxylipins could mediate the effect of fried foods on CAD risk was included.
  2. Results: Table 2 shows how small the differences between upper and lower limits of the middle tertile (0.004 - 0.7 ppb). How significant is it compared to the variance across technical replicates used for each sample (page 4 line 125-127)? Will it be possible for the authors to provide the mean and confidence interval of oxylipin levels from each tertiles?
  3. Results: Figure 2 shows how adding 4 oxylipins improve the prediction of CAD incidence. Have the authors considered adding the oxylipin one by one if the prediction is improved by adding only a single oxylipin, or which combination of oxylipins will give the best prediction?
  4. Have the authors considered to validate the model in an independent subset? Or to divide the subset to training and validation subsets to validate the model?

Specific comments:

  1. Methods
    1. Page 2 line 70-72 section 2.1. Research design and participants: baseline information on the blood cholesterols (i.e., VLDL, LDL, HDL, and total cholesterols) was not listed although they were included in Table 1
    2. Page 3 Figure 1: what is the cut-off limit used to exclude the outliers of oxylipins? It would be good to specify that in the text

Reviewer 3 Report

This study aims at investigating whether serum oxylipin profile is associated with the risk of developing CAD and whether including identified oxylipins may improve the predictability of CAD risk. The manuscript is well-written, clearly presented and well organized. It brings very relevant information in the field of oxylipins and cardiovascular health.

Major concerns are related to experimental and/or theoretical methods including biostatistics we should be revised before publication.

  1. In the Research design section, could you please precise how were defined the Case participants (time before the first CAD event, number of CAD..etc).
  2. In the Data processing section, it is not clear what are the three replicates? Did the authors used three different serum samples and replicate the oxylipin extraction?
  3. In the statistical analysis section
    1. How the qualitative data (e.g. smoking, education level) were compared in Table 1? Fisher test is usually recommended.
    2. The relevance of adjusting the conditional logistic regression models with so many factors involved in the pathogenesis of CAD (i.e. hypertension, diabetes..) is highly questionable. By doing so, this exclude a large number of oxylipins that could be predictive of CAD and interesting to finely phenotype the Case participants. The authors should consider adjusting only on factors that are not related to the pathogenesis, at least as a first option.
    3. Conditional logistic regression models are a good choice but these are not well fitted with multivariate dataset. The authors should apply a penalization (e.g. elastic-net, Lasso) to avoid over-fitting their models.

Minor corrections:

  1. The Introduction should be revised as it relates a lot to nutrition and frying which is not in line with the presented results. I would recommend to explain the rational of focusing on oxylipins to predict CAD.
  2. The Introduction should be better referenced

Round 2

Reviewer 1 Report

The authors revised the manuscript substantially with adding supplementary data. These changes are mostly favorable, however, this reviewer still wonder how the authors select the four oxylipins, but not the others, as candidates of CVD-related parameters. I think further explanation for the process of selecting the parameters and scientific significance would be beneficial for the readers.

Major points

  1. Thank you for adding the amounts of all the oxylipin molecules detected as Supplementary Table S2 which is very informative to overview this study. According to the data in Table S2, the concentrations of all the oxylipin species detected are very constant and I could not see any difference between cases and controls. It is hard to imagine how the authors picked up only 4 oxylipin species from these data. For example, the concentrations pf PGE2/PGD2 in controls and in cases are 0.00100000 +/- 0.00000021 and 0.00099999 +/- 0.00000026, respectively. It means the mean concentration in cases is 99.999% of the control, and the standard deviation is less than 0.03 % of the mean value. Thus, the concentration of PGE2/PGD2 is very constant in the cohort and the deviation is very limited. I cannot understand such a constant parameter could be a useful for determining the CVD cases. Additional explanation for the concept of data processing and the detailed calculation methodologies would be helpful.

Minor points

  1. The top line of the Table S2. The Control (N=77) should be Cases (N=77).
  2. The unit used in Table 2, Table 3 and Table S3 is ppm, while the unit in Table S3 is ppb.
  3. I would like to suggest the authors should consider “significant digits” of the data.
